# Origin of Ferroelectricity in BiFeO_3_-Based Solid Solutions

**DOI:** 10.3390/nano12234163

**Published:** 2022-11-24

**Authors:** Yuji Noguchi, Hiroki Matsuo

**Affiliations:** 1Division of Information and Energy, Faculty of Advanced Science and Technology, Kumamoto University, 2-39-1, Kurokami, Chuo-ku, Kumamoto 860-8555, Japan; 2International Research Organization for Advanced Science & Technology (IROAST), Kumamoto University, 2-39-1, Kurokami, Chuo-ku, Kumamoto 860-8555, Japan

**Keywords:** perovskite, ferroelectric, polarization, BiFeO_3_, solid solution, hybridization, Bloch function, rhombohedral, tetragonal, orthorhombic

## Abstract

We investigate the origin of ferroelectricity in the BiFeO_3_–LaFeO_3_ system in rhombohedral *R*3*c* and tetragonal *P*4*mm* symmetries by ab initio density functional theory calculations and compare their electronic features with paraelectric orthorhombic *Pnma* symmetry. We show that a coherent accommodation of stereo-active lone pair electrons of Bi is the detrimental factor of ferroelectricity. A Bloch function arising from an indirect Bi_6*p*–Fe_3*d* hybridization mediated through O_2*p* is the primary origin of spontaneous polarization (*P*_s_) in the rhombohedral system. In the orthorhombic system, a similar Bloch function was found, whereas a staggered accommodation of stereo-active lone pair electrons of Bi exclusively results in paraelectricity. A giant *P*_s_ reported in the tetragonal system originates from an orbital hybridization of Bi_6*p* and O_2*p*, where Fe-3*d* plays a minor role. The *P*_s_ in the rhombohedral system decreases with increasing La content, while that in the tetragonal system displays a discontinuous drop at a certain La content. We discuss the electronic factors affecting the *P*_s_ evolutions with La content.

## 1. Introduction

Perovskite ferroelectrics have attracted much attention because of their fascinating properties, such as electromechanical response and switchable spontaneous polarization (*P*_s_) by external stimuli. Lead titanate (PbTiO_3_) has tetragonal *P*4*mm* symmetry and features a high Curie temperature (*T*_C_) of 495 °C and a robust *P*_s_ [1]. Ferroelectric tetragonal PbTiO_3_ forms a solid solution with antiferroelectric rhombohedral PbZrO_3_ [2,3]. Their dielectric and piezoelectric properties are maximal at around the morphotropic phase boundary (MPB) [4], and the properties around the MPB have been widely utilized for sensors, ultrasonic motors, and medical transducers [5,6,7]. Single crystals of Pb(Mg, Nb)O_3_–PbTiO_3_ and Pb(Zn, Nb)O_3_–PbTiO_3_ exhibit extraordinary high electromechanical coupling factors because a bridging phase in monoclinic symmetry [4,8,9,10] is capable of a rotation of *P*_s_ under electric fields [8,11,12,13,14].

Ferroelectric BiFeO_3_ has been intensively studied from the viewpoints not only of multiferroic nature [15,16,17,18] but also of a Pb-free piezoelectric material [19,20,21,22]. In BiFeO_3_ [18,23], *P*_s_ coexists with an incommensurate spin structure, which can be approximated as an antiferromagnetic spin configuration [15,17]. A large *P*_s_ of 80–100 μC/cm^2^ along with a high *T*_C_ of 830 °C [24,25] is present along [111]_c_ in rhombohedral *R*3*c* symmetry [24,25] (the subscript ‘c’ indicates pseudo-cubic), while strained tetragonal films display a giant *P*_s_ of over 130 μC/cm^2^ [26].

A formation of composition-driven phase boundaries, such as MPBs, is expected in the systems of BiFeO_3_–REFeO_3_ [27,28,29,30,31,32,33,34,35] (RE: rare-earth elements) and BiFeO_3_–BaTiO_3_ [20,21,36,37]. The simplest solid solution is the BiFeO_3_–LaFeO_3_ system with a composition of Bi_1−*x*_La*_x_*FeO_3_, the detail of which has been comprehensively summarized in the review paper [27]. Karimi et al. [28] reported that a phase boundary between the ferroelectric *R*3*c* and the paraelectric orthorhombic *Pnma* phases exists at *x* ~ 0.23 at room temperature. Rusakov et al. reported that single-phase materials with *R*3*c* symmetry can be prepared after annealing for composition 0 ≤ *x* ≤ 0.1, and the *Pnma* phase is stable at 0.50 ≤ *x* ≤ 1; these results were verified by synchrotron radiation X-ray diffraction, electron diffraction, and high-resolution transmission electron microscopy [38]. They also found that an incommensurate phase in orthorhombic *Imma* symmetry is formed at 0.19 ≤ *x* ≤ 0.30. Karpinsky et al. [29] proposed a temperature-composition phase diagram, in which the ferroelectric *R*3*c* (*x* < 0.15) and the paraelectric *Pnma* (*x* > 0.4) phases are mediated through a bridging anti-polar phase in orthorhombic *Pbam* symmetry.

One possible reason for the wide variety of experimental reports on the phase diagrams is incomplete solubility of La on the A site [39]. Moreover, the abovementioned phases are energetically competing with each other and some of them are likely to be energetically degenerate [31]. Therefore, ab initio studies based on density functional theory (DFT) are expected to provide clues for uncovering the ground-state crystal structure and the phase stability.

Lee et al. investigated the effect of the La doping on the variation of the off-center distortion and the orbital mixing in BiFeO_3_ by experiments in conjunction with DFT calculations [30]. They reported that both an Fe–O bond anisotropy and off-center cation displacements are suppressed by the La doping. As a result, the degree of Fe 3*d*–4*p* orbital mixing decreases in the solid solution samples. An impact of the La content on the polarization and the electronic band structure was also reported by You et al. [40]. They reported that the La doping induces a chemically driven rotational instability. It modifies the local crystal field along with the electronic structure, which gives rise to a direct-to-indirect transition of the bandgap and provides an enhancement in ferroelectric photovoltaic effect. In contrast, Tan et al. reported that the La doping has little influence on *P*_s_ in tetragonal BiFeO_3_ [34]. In spite of extensive research by DFT studies [30,31,32,33,34,35], the *P*_s_’s evolution with the La content and its electronic origin still remain unclear.

The purpose of this study is to elucidate the origin of ferroelectricity in rhombohedral *R*3*c*, tetragonal *P*4*mm* in the BiFeO_3_-LaFeO_3_ system (Bi_1−*x*_La*_x_*FeO_3_). The electronic feature and structural distortion in the paraelectric orthorhombic *Pnma* symmetry are also investigated because Bi_1−*x*_La*_x_*FeO_3_ with *x* ≥ 0.5 is of the orthorhombic phase [29]. We show DFT energy evolutions with *x* but focus on the relation between the orbital hybridizations and the ferroelectric (paraelectric) distortions. We show that the Bloch function arising from a Bi_6*p*-Fe_3*d* hybridization mediated through O_2*p* is the primary origin of *P*_s_ in the rhombohedral system. In the orthorhombic system, a similar Bloch function and the resultant structural distortion are constructed, whereas a staggered accommodation of stereo-active lone pair electrons of Bi never allow the presence of *P*_s_. We discuss a large *P*_s_ and its dependence on *x* in the tetragonal system.

## 2. DFT Calculations

Density functional theory (DFT) calculations were conducted using the generalized gradient approximation [41] with a plane-wave basis set. We used the projector-augmented wave method [42] as implemented in the Vienna ab initio simulation package (VASP) [43]. We employed the Perdew–Burke–Ernzerhof gradient-corrected exchange-correlation functional revised for solids (PBEsol) [44], a plane-wave cut-off energy of 520 eV, an electronic iterations convergence of 1 × 10^−6^ eV, and a criterion for ionic relaxations of 1 meV/nm. The Γ-centered *k*-point mesh was set to 3 × 3 × 3 for the structural optimizations and 5 × 5 × 5 for density of states (DOS) and band structure calculations.

Within the simplified generalized gradient approximation (GGA) + *U* approach [45], we added on-site Coulomb interaction parameters of *U*–*J* of 6 eV to Fe-3*d* throughout the calculations. The on-site Coulomb interaction parameters of *U–J* for Fe-3*d* has been employed in the range of 2–6 eV for BiFeO_3_ [34,45,46,47]. The bandgap value is enlarged when *U–J* is increased, while the essential feature, such as *P*_s_, and the valence-band electronic structure remain unchanged. One main reason why we adopted *U–J* = 6 eV is as follows: the bandgap becomes narrow for a specific Bi–La arrangement on the A site and eventually vanishes when the arrangement of Bi and La is an ordered configuration along the polar *c* axis, as will be described later. In order to maintain the bandgap above ca. 2 eV, we set *U–J* to 6 eV for Fe-3*d* throughout the calculations.

Considering the spin configuration in BiFeO_3_ can be approximated as the *G*-type antiferromagnet [46], we set the spin arrangement in which the adjacent Fe ions have an antiparallel spin configuration as much as possible irrespective of the La content (*x*) on the A site.

We employed three symmetries shown in Figure 1 and compared their total energies (see Figure 2): ferroelectric rhombohedral *R*3*c*, ferroelectric tetragonal *P*4*mm*, and nonpolar orthorhombic *Pnma*. The antiparallel spin configuration of adjacent Fe ions enforces a change in space group from *R*3*c* to *R*3 (or *P*3) for the rhombohedral cells, from *Pnma* to *P*2_1_/*m* for the orthorhombic ones, while the tetragonal *P*4*mm* remains unchanged, the details of which are summarized in Appendix A. Although the orthorhombic cells were optimized in monoclinic *P*2_1_/*m* symmetry, we restricted the monoclinic angle of *β* to 90.0 degree, and then the crystal system is regarded as orthorhombic.

Because our calculations were performed in the anti-ferromagnetic spin configuration, the total DOS is identical both in the majority and minority spin bands. In a similar manner, the band structure in the majority spin band is the same with that in the minority spin band. We display the DOS in the majority spin band in the right panel and that in the minority spin band in the left one, e.g., see Figure 3c. For simplicity, we show the band structure only in the majority spin band. As for wavefunctions (partial charge densities), we visualize the majority spin component at an iso-surface level of 1 × 10^−4^, e.g., see Figure 4. The Fermi energy is set to zero throughout our manuscript.

For building a BiFeO_3_–LaFeO_3_ solid solution cell, there exists several choices of the arrangement of Bi and La. We adopted the rock-salt arrangement of Bi and La, especially along the polar *c* axis, as much as possible, as can be seen in Appendix A. This is because the electronic structure in the rhombohedral cell with a layered Bi and La ordering along the polar ***c*** axis has a non-realistic metallic feature (Appendix A), which is not consistent with the experimental fact of an insulating nature of BiFeO_3_–LaFeO_3_ solid solutions [47,48]. We therefore avoid such a Bi–La ordering along the specific crystallographic axis and adopt the rock-salt-like orderings of Bi and La. The details of the structure parameters are listed in Appendix A for the rhombohedral cells, Appendix A for the tetragonal cells, and Appendix A for the orthorhombic cells.

From the structure parameters of the optimized cell (Appendix A), we obtained the atomic displacements (∆*z*) from the corresponding positions in the hypothetical non-polar paraelectric lattice. We also calculated the Born effective charges (*Z**) [49] by density-functional perturbation theory. We estimated spontaneous polarization (*P*_s_) using the following equation
(1)$$P_{s}=\sum_{i}m_{i}\cdot \Delta z_{i}\cdot Z_{i}^{*}/V,$$
where $m_{i}$ denotes the site multiplicity of the constituent atom *i* and $\Delta z_{i}\cdot Z_{i}^{*}$ its dipole moment. The summation in Equation (1) is taken over the cell with cell volume (*V*).

## 3. Results and Discussion

### 3.1. Ground-State Structures

Figure 2a shows the total energy (*E*_total_) per the ABO_3_ formula unit as a function of the La content (*x*) expressed by *x* = [La]/([Bi] + [La]). For comparing *E*_total_ in different symmetries, *x* should be the same. Indeed, *x* has a discrete value depending on the total number of the A-site atoms, i.e., six in the rhombohedral cells, eight in the tetragonal ones, and four in the orthorhombic ones (Appendix A). We, therefore, set the energies obtained by fitting the DFT energies of the rhombohedral cells by a quadratic function to zero of *E*_total_. The *E*_total_ of the rhombohedral cells are lower than those of the tetragonal and orthorhombic cells irrespective of *x*. These results are different from those reported in the literature [31,50,51]. The orthorhombic cells are slightly higher in *E*_total_ by ~0.1 eV while the tetragonal cells have a higher *E*_total_ by ~0.3 eV. In reality, the tetragonal BiFeO_3_ is stabilized under a compressive strain in epitaxial films [26].

Figure 2b shows the *P*_s_ of the rhombohedral cells as a function of *x*. The *P*_s_ of the rhombohedral BiFeO_3_ is 83.2 μC/cm^2^, which is in good agreement with the previous DFT calculations [26,31,52] and experiments [53]. With increasing *x*, the rhombohedral *P*_s_ exhibits a monotonic decrease, which is consistent with other calculations [31]. The tetragonal BiFeO_3_ (Figure 2c) possesses a giant *P*_s_ of 133.4 μC/cm^2^, which is close to the values obtained for epitaxially strained films and from DFT calculations [26]. Zhang et al. [26] have reported that this large *P*_s_ is associated with a high tetragonality *c*/*a* of 1.24 and also with a coherent displacement of Fe (∆_Fe_’) by 0.033 nm with respect to the Bi sublattice. These values accord with our calculations of *c*/*a* = 1.25 and ∆_Fe_’ = 0.032. The *P*_s_ and *c*/*a* decrease with increasing *x* followed by discontinuous drops at 2/8 < *x* < 3/8. At *x* = 3/8, the *c*/*a* is almost in unity whereas the apparent *P*_s_ of 52.0 μC/cm^2^ exists. With a further increase in *x*, the *P*_s_ again shows a monotonic decrease while the *c*/*a* remains constant (ca. ~1.0).

### 3.2. Electronic Structures

#### 3.2.1. Ferroelectric Rhombohedral System

Figure 3 shows (a,b) the crystal structures of the rhombohedral BiFeO_3_ (*x* = 0) along with (c–g) the total and partial DOS results and (h,i) the electronic band structures in the valence band. The wavefunction of the band shown in the blue circle in i is depicted in Figure 4. The off-center displacements of Fe and Bi are stabilized by the following two orbital hybridizations, respectively: Fe_3*d–*O_2*p* (Figure 5a) and Bi_6*p*–O_2*p* (Figure 5b). Because Fe atoms have either a positive (↑) or a negative (↓) magnetic moment, the majority spin (↑) and minority spin (↓) bands have to be taken into account in a distinct manner. Here, we consider the orbital interaction in the ↑ band that leads to DOS components in the valence band in the range of −7 to 0 eV (in the right panel in Figure 3c–g). Although O1 and O2 have a slightly different magnetic moment, the DOS characters are almost identical both in the ↑ and ↓ bands. When the magnetic moments are ignored, O1 and O2 have the same site symmetry, and therefore, we do not distinguish O1 and O2. Due to the same reason, we regard Bi1 and Bi2 as identical Bi atoms. 

The simple ionic model considering the electron configuration of 6*s*^2^ 6*p*^0^ for Bi^3+^ leads to zero DOS of Bi_6*p* in the valence band, because the states of the isolated Bi_6*p* are unoccupied. The Bi_6*p*–O_2*p* interaction (Figure 5b) results in a low-lying bonding state and a high-lying antibonding state. The low-lying states are occupied by electrons and thereby the Bi_6*p* states have apparent DOS components in the valence band (Figure 3e).

Next, we consider the Fe_3*d*–O_2*p* interaction leading to the DOS component in the ↑ band. The Fe1 atom with a negative magnetic moment has the electron configuration of 3*d* with ↓^5^↑^0^. These five ↓ electrons of Fe1 are present at deep levels in the ↓ band [54]. We focus our attention on the interaction between the empty Fe1_3*d* (↑) and the occupied O_2*p* states (Figure 5a). The hybridization of these orbitals delivers an occupied bonding state and an empty antibonding state. Therefore, Fe1 has not only the major DOS in the band but also the minor DOS in the ↑ band, as displayed in Figure 3d.

The bands in the range of −7 to −5 eV (Figure 3h) are derived primarily from the bonding states of Fe2_3*d* (↑)–O_2*p*. The Bi_6*p*–O_2*p* interaction yields a low-lying bonding state at around −4.5 eV at the Γ point (marked with a blue circle in Figure 3i), whose wavefunction is visualized in Figure 4. The distinct Fe1_3*d–*O_2*p* and Bi_6*p*–O_2*p* interactions are seen in the respective local regions of Fe1O_6_ octahedra and BiO polyhedrons. We note that the interaction between the bonding states of Fe1_3*d–*O_2*p* and Bi_6*p*–O_2*p* (Figure 5c) forms a coherent wavefunction that is spread throughout the crystal (Figure 4). Namely, the interaction between the Fe1_3*d–*O_2*p-* and the Bi_6*p*–O_2*p*-derived bonding states yields the low-lying bonding state, termed Bloch function, arising from the –Fe1–O–Bi–O– network (Figure 4). It results also in the high-lying antibonding state in the valence band, as shown in Figure 5c. We conclude that the Bloch function stemming from the indirect Bi_6*p*–Fe_3*d* hybridization mediated through O_2*p* is the primary origin of *P*_s_ in the rhombohedral system.

Appendix A shows (a) the crystal structure of the rhombohedral cell (*x* = 2/6) along with (b–f) the total and partial DOS results and (g,h) the electronic band structures in the valence band. The wavefunction of the band shown in the blue circle in h is depicted in i. The orbital interactions displayed in Figure 4 are seen also in this cell. The Bloch function arising from the Fe_3*d*–O_2*p*–Bi_6*p* hybridization appears at −4.5 eV (at the Γ point), while its connection along the *c* axis is relatively weak compared with that in the BiFeO_3_ cell. The similar electronic feature was found for the rhombohedral cell (*x* = 4/6) in Appendix A. It is interesting to note that the Bloch function is formed through the Bi–O–Fe bond avoiding La.

#### 3.2.2. Ferroelectric Tetragonal System

The large *P*_s_ of the tetragonal BiFeO_3_ is derived from the cooperative off-center displacements of Bi and Fe along the polar *c* axis with respect to the oxygen sublattice (Appendix A for the BiFeO_3_ cell). This off-center feature is maintained in the tetragonal cell with *x* = 2/8. Appendix A shows (a) the crystal structure of the tetragonal cell (*x* = 2/8) along with (b–f) the total and partial DOS results and (g, h) the electronic band structures in the valence band. The wavefunction of the band shown in the blue circle in h is depicted in i. The Bloch function where a Bi_6*p*–O2_*p* interaction plays a central role (Appendix A) is present at ca. −4.2 eV at the Γ point and spreads also along the *c* axis through the Bi_6*p_z_* orbital. Note that the Fe-3*d* state indeed does not participate in the Bloch function, which is in contrast to the rhombohedral system (Figure 4, Appendix A). Additionally, in the BiFeO_3_ cell (Appendix A), the similar Bloch function with a small contribution of Fe-3*d* appears.

Figure 6 shows (a) the crystal structure of the tetragonal cell (*x* = 3/8) along with (b–f) the total and partial DOS results and (g,h) the electronic band structures in the valence band. The wavefunction of the band shown in the blue circle in h is depicted in (i). The Bi_6*p*–O2_*p* hybridization results in a bonding state at ca. −4.6 eV at the Γ point. The Bloch function has a robust connection in the *a*–*a* plane whereas that is markedly weakened along the polar *c* axis. The discontinuous drops in *P*_s_ and *c*/*a* at 2/8 < *x* < 3/8 stems from the in-plane feature of the Bloch function formed by the Bi_6*p*–O2_*p* hybridization.

#### 3.2.3. Paraelectric Orthorhombic System

We think that the variations of *P*_s_ and *c*/*a* can be qualitatively understood from a decrease in the number of the Bi pillar along the ***c*** axis. Appendix A shows the arrangement of Bi and La in the tetragonal cells at (a) *x* = 2/8 and (b) *x* = 3/8. The tetragonal BiFeO_3_ (*x* = 0) has a large *P*_s_, which is ascribed to the full set of the Bi pillars. The substitution of La on the A site decreases the number of the Bi pillar and two Bi pillars are maintained until *x* ≤ 2/8. These Bi pillars contribute to the formation of the Bloch function spreading along the *c* axis through the Bi_6*p_z_* orbital. With increasing *x* above 3/8, the number of the pillar becomes only one, and thereby the Bloch function has an in-plane feature, which is accompanied by a marked decrease in *P*_s_.

Figure 7 shows (a) the crystal structure of the orthorhombic cell (*x* = 1/2) along with (b–e) the total and partial DOS results and (f,g) the electronic band structures in the valence band. The wavefunction of the band shown in the blue circle (g) is depicted in (h). The magnetic moment of Fe1 is negative, and then the orbital interaction shown in Figure 6a is expected for Fe1 and the adjacent oxygen atoms. The Fe1_3*d* states have a marked DOS component in the ↑ band, which stems from the orbital hybridization of Fe1_3*d* (↑)–O3_2*p*. An apparent DOS of Bi1_6*p* arising from the mixing with O3_2*p* appears in the valence band. The wavefunction of the band at the Γ point (Figure 7h) clearly shows that the Bloch function originating from a coherent interaction between the bonding states of Fe1_3*d* (↑)–O3_2*p* and Bi1_6*p*–O3_2*p* spreads throughout the crystal. Moreover, this Bloch function is present at −5.4 eV, which is lower by 0.9 eV than that of the rhombohedral cell (*x* = 2/6) (Figure 3).

### 3.3. Factors Affecting Ferroelectricity

As described above, the Bloch function arising from the indirect Bi_6*p*–Fe_3*d* hybridization via O_2*p* is the origin of *P*_s_ in the rhombohedral system. The hybridized orbital is accompanied by the formation of the covalent bonds not only in the direction of *P*_s_ but also in the opposite direction of *P*_s_, e.g., see Figure 4. The Bloch function is closely related to the bond lengths of Bi–Fe (see Appendix A); this is the first (structural) factor leading to a robust *P*_s_.

In the rhombohedral system (Appendix A), the first shortest Bi–Fe bond is parallel to *P*_s_, while the second shortest Bi–Fe bond is almost normal to *P*_s_. For the tetragonal system (Appendix A), the first and second shortest Bi–Fe bonds are aligned along pseudo-cubic <111> direction. In the rhombohedral cells at *x* ≤ 4/6, the first shortest Bi–Fe length is ~0.31 nm, and the second shortest one is ~0.33 nm. In the tetragonal cells, the first shortest Bi–Fe length is ~0.33 nm, which is comparable to that in the rhombohedral one. We note that the second shortest length is as long as ~0.37 nm even at *x* = 0 (BiFeO_3_). This long Bi–Fe length does not allow the Fe_3*d* orbital to participate in the Bi_6*p*–O_2*p* hybridization, and thereby, the Bloch function is indeed formed by the Bi_6*p*–O_2*p* interaction in the tetragonal system, as shown in Figure 6.

In the paraelectric orthorhombic cells at *x* ≤ 1/2, the first and second shortest Bi–Fe lengths are ~ 0.32 nm and ~ 0.33 nm, respectively, which are comparable to those in the rhombohedral cells. Actually, the Bloch function resulting from the Fe_3*d*–O_2*p*–Bi_6*p* mixture is extended throughout the crystal at *x* = 1/2 (see Figure 7). The most significant difference between the ferroelectric and paraelectric cells appears in an accommodation mode of stereo-active lone pair electrons of Bi [26,55,56] (derived from the mixture of Bi_6*s*–6*p*), which is the second factor that dominates the presence or absence of *P*_s_. For the ferroelectrics in the rhombohedral (Figure 4a) and tetragonal systems (Figure 6i), the lone pair electrons of Bi are directed coherently along the *c* axis, which is the detrimental factor of ferroelectricity. The alignment of the lone pair electrons contributes to an enhancement of off-center displacements of cations, as in ferroelectric PbTiO_3_ [57]. For the orthorhombic cell (Figure 7h), the lone pair electrons of Bi1 (*y* = 0.75) are directed opposite to those of Bi1 (*y* = 0.25) along the *c* axis (which is the symmetry constraint). This staggered accommodation of the lone pair electrons of Bi exclusively provides paraelectricity.

The Bi–Fe bond lengths and the resultant Bloch functions can be qualitatively understood by the tilt and rotation of FeO_6_ octahedra expressed by the Glazer notation [58]. The ferroelectric rhombohedral cells have an out-of-phase octahedral tilt expressed by $a^{-}a^{-}a^{-}$, which are accompanied by the short Bi–Fe bonds along with relatively large unit-cell densities of 8.595 g cm^−3^ at *x* = 0 and 7.779 g cm^−3^ at *x* = 0.5. In contrast, the ferroelectric tetragonal cells do not have any tilt or rotation of FeO_6_ octahedra at *x* = 0 and 0.5, leading to the long (second shortest) Bi–Fe lengths and low densities of 8.027 g cm^−3^ at *x* = 0 and 7.669 g cm^−3^ at *x* = 0.5. The paraelectric orthorhombic cells have a tilt system of $a^{-}b^{+}a^{-}$, resulting in the short Bi–Fe lengths and high densities of 8.836 g cm^−3^ at *x* = 0 and 7.807 g cm^−3^ at *x* = 0.5.

## 4. Conclusions

We have investigated the origin of ferroelectricity of Bi_1−*x*_La*_x_*FeO_3_ in rhombohedral *R*3*c* and tetragonal *P*4*mm* symmetries by DFT calculations. In the rhombohedral system, a Bloch function arising from an indirect Bi_6*p*–Fe_3*d* hybridization via O_2*p* is the primary origin of *P*_s_. In contrast, the *P*_s_ of the tetragonal phase stems from a Bloch function arising from a Bi_6*p*–O_2*p* mixing with a weak contribution of Fe-3*d*. The detrimental factor of the presence/absence of *P*_s_ is an accommodation of stereo-active lone pair electrons of Bi. The paraelectric orthorhombic *Pnma* phase has a staggered accommodation of lone pair electrons of Bi, while the ferroelectric *R*3*c* and *P*4*mm* systems exhibit a coherent alignment of lone pair electrons of Bi. The rhombohedral system shows a monotonic decrease in *P*_s_ with increasing *x*, which is directly associated with a weakening of the Fe_3*d*–O_2*p*–Bi_6*p* hybridization. In contrast, the tetragonal system displays a discontinuous drop of *P*_s_ at ca. *x* = 0.3, which is ascribed to a transition from a 3D extension to an in-plane feature of the Bi_6*p*–O_2*p* mixed orbital.

## Figures and Tables

**Figure 1 nanomaterials-12-04163-f001:**
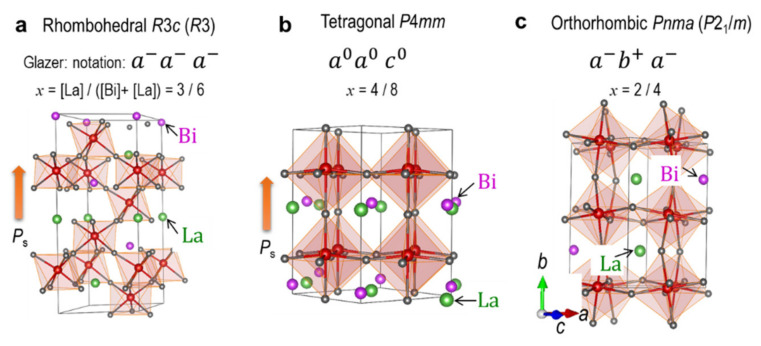
Optimized crystal structures in (**a**) rhombohedral, (**b**) tetragonal, and (**c**) orthorhombic symmetries of the Bi_1−*x*_La*_x_*FeO_3_ cells with *x* = 1/2. The tetragonal cells with *x* = 0, ½, and 1 do not have octahedral tilt, while those with *x* = 1/8, 2/8, 3/8, 5/8, and 7/8 have their distinct tilt modes. We adopt a rock-salt structure in an arrangement of Bi and La on the A site as much as possible, the details of which are displayed in Appendix A. The symmetry in parenthesis denotes the space group taking account of the antiferromagnetic spin configuration.

**Figure 2 nanomaterials-12-04163-f002:**
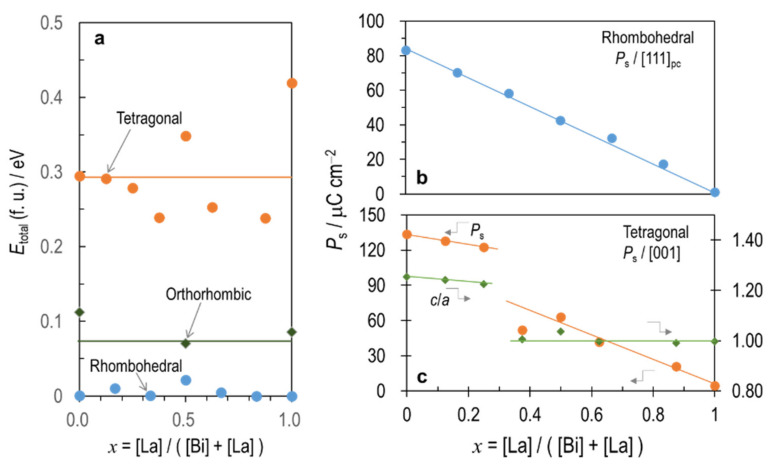
Phase stability and *P*_s_: (**a**) total energy (*E*_total_) of the ABO_3_ formula unit (f. u.); (**b**) *P*_s_ in the rhombohedral system; and (**c**) *P*_s_ and *c*/*a* in the tetragonal system as a function of La content (*x*). For comparing *E*_total_ in different symmetries, we set the energies obtained by fitting the DFT energies of the rhombohedral cells by a quadratic function to zero.

**Figure 3 nanomaterials-12-04163-f003:**
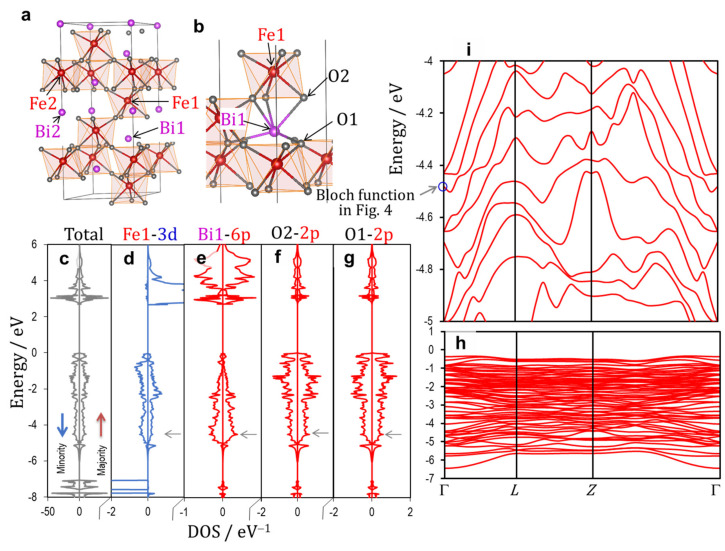
Crystal structures (**a**,**b**), electronic density of states (DOS) (**c**–**g**), and band structures (of the majority spin band) in the valence band (**h**,**i**) of the rhombohedral BiFeO_3_ (*x* = 0). The wavefunction of the band shown in blue circle in (**i**) is displayed in Figure 4. The up (red) and down (blue) arrows in (**c**) denote the majority and minority spin bands, respectively. The horizontal arrows (gray) in (**c**–**g**) correspond to the energy level of the blue circle in (**i**).

**Figure 4 nanomaterials-12-04163-f004:**
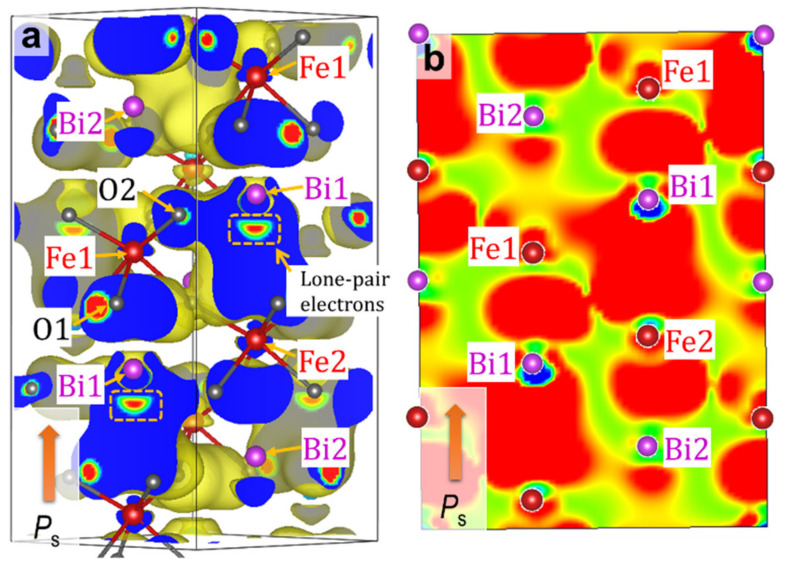
Wavefunction (partial charge density) of the ↑ band shown in the blue circle in (**i**) of the rhombohedral BiFeO_3_ (*x* = 0): (**a**) 3D plot and (**b**) 2D visualization on the lattice plane including Bi1 and Fe1. This wavefunction arising from an indirect Bi1_6*p*–Fe1_3*d* hybridization mediated through O_2*p* is termed ‘Bloch function’. The roles of O1 and O2 in this band are almost identical, and therefore, we do not distinguish them.

**Figure 5 nanomaterials-12-04163-f005:**
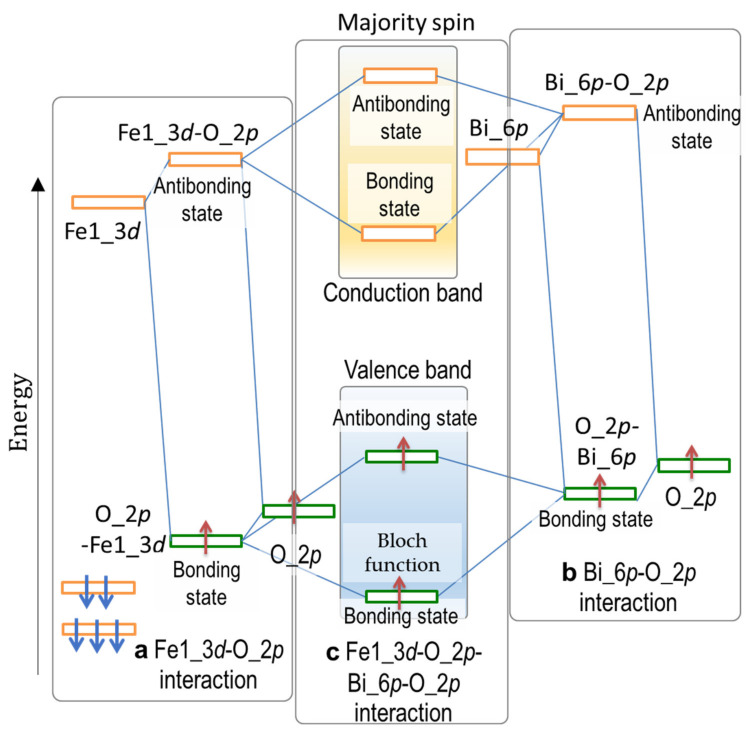
Orbital interactions delivering the Bloch function of the majoring spin (↑) band shown in Figure 4 of the rhombohedral BiFeO_3_ (*x* = 0): (**a**) Fe1_3*d*–O_2*p* mixing leading to a bonding state in the valence band and an antibonding state in the conduction band; (**b**) Bi_6*p*–O_2*p* mixing resulting in a low-lying bonding state and a high-lying antibonding state; and (**c**) hybridization between Fe1_3*d*–O_2*p* and Bi_6*p*–O_2*p* leading to the Bloch function at a deep level in the valence band.

**Figure 6 nanomaterials-12-04163-f006:**
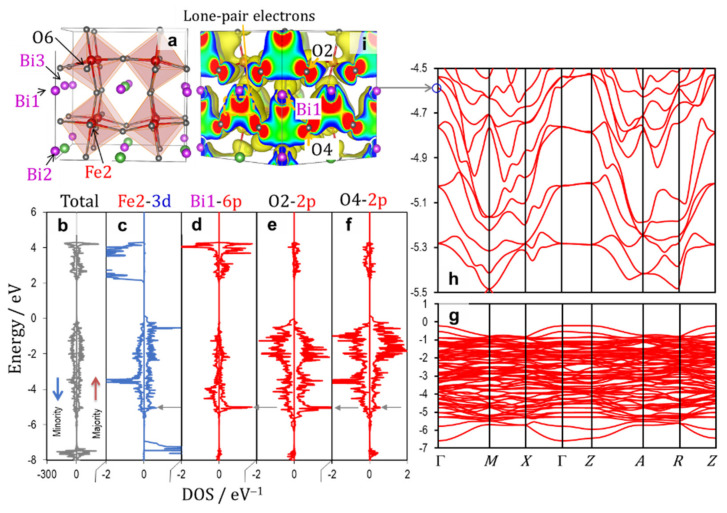
Crystal structure (**a**), electronic density of states (DOS) (**b**–**f**), band structures (of the majority spin band) in the valence band (**g**,**h**), and **i** wavefunction of the band shown in the blue circle of the tetragonal cell (*x* = 3/8). The up (red) and down (blue) arrows in (**b**) denote the majority and minority spin bands, respectively. The horizontal arrows (gray) in (**c**–**g**) correspond to the energy level of the blue circle in (**h**).

**Figure 7 nanomaterials-12-04163-f007:**
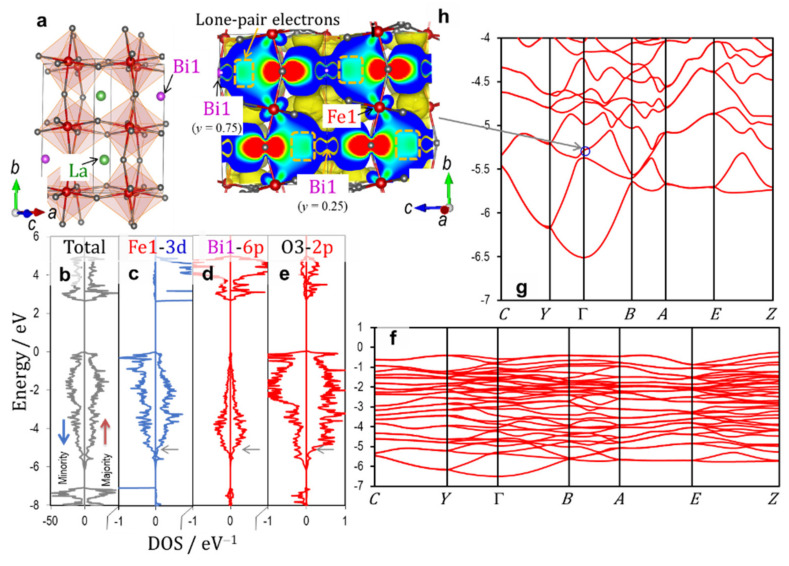
Crystal structure (**a**), electronic density of states (DOS) (**b**–**e**), band structures (of the majority spin band) in the valence band (**f**,**g**), and (**h**) wavefunction of the band shown in the blue circle of the orthorhombic cell (*x* = 1/2). The up (red) and down (blue) arrows in (**c**) denote the majority and minority spin bands, respectively. The horizontal arrows (gray) in (**c**–**e**) correspond to the energy level of the blue circle in (**g**).

## Data Availability

The data that support the findings of this study are available upon reasonable request from the corresponding author.

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
