# Peer review of "Origin of Ferroelectricity in BiFeO3-Based Solid Solutions"

_nanomaterials, 2022, doi:10.3390/nano12234163_

Round 1

Reviewer 1 Report

In the present manuscript the authors describe the results of density functional theory modelling of the structure and electronic states of BiFeO3 (BFO) as well as solid solutions of BFO with different amounts of LaFeO3 (LFO), considering both ferroelectric rhombohedral and tetragonal phases as well as the paraelectric orthorhombic phase. The origin of ferroelectricity in the rhombohedral phase is traced mainly to the hybridization of Bi_6p and Fe_3d states via the 2p states of the oxygen connecting the two. In the tetragonal phase, chains of Bi-ions create high polarization, while the contribution of Fe ions is reduced.

The findings are presented well and relatively easy to understand, though the manuscript suffers from a high number of typographical errors and would benefit from additional proof reading. The topic is of high interest, and the results presented are also sufficiently novel and embedded in existing literature. It also appears to be mostly scientifically sound, though there are a few points concerning both style and content that should still be addressed:

- For the calculations a rock-salt ordering of Bi and La in the solid solution was chosen. What is the reason for this? Were similar calculations carried out for other ionic arrangements? Did the type of ordering have a notable influence on the results? There should at least be a statement on this point included in the manuscript.

- The blue circle marking the relevant state in Fig. 3 i) is very hard to make out. In the other figures, it was additionally marked with an arrow; could this be added in Fig. 3 too?

- What is the point in showing all the bands in Fig. 3 h), 6 g) and 7 f)? Due to the high density of the lines, it is not possible to see any details. What information is the reader supposed to draw from this? Could these parts of the figure be left out?

- The words ‘Bonding state’, ‘Antibonding state’ etc. in Fig. 5 are very hard to make out. Could this be presented in a larger font?

- The results show that the rhombohedral phase is the only stable phase across the entire composition range, and the tetragonal phase is energetically the most unfavorable in the unstressed state. As the authors point out, mechanical stress would stabilize the tetragonal phase. For the real-world understanding of the behavior of BLFO, it would therefore be interesting to know how this stress influences the results concerning the origin of ferroelectricity. Can it be assumed that the information on the hybridization of Bi_6p and O_2p being a major carrier of the ferroelectric polarization still holds true in the stressed system?

- The authors note that the drop in polarization in the tetragonal phase between x = 0.25 and x = 0.375 is due to the reduction of the number of Bi-chains through the structure. However, the threshold value for this should again depend on the Bi/La ordering. If the order is changed to allow for more Bi-chains at higher La content, does the threshold for the drop in polarization also shift?

If these points can be clarified or amended in the manuscript, this work can be a notable contribution to the understanding of ferroelectricity in BFO-based materials.

Author Response

Dear reviewers,

Thank you very much for your useful and constructive comments on our manuscript. I apologies for many typo errors. We have completely removed these errors in our revise manuscript. In addition, our replies on the comments are listed below.

Yuji Noguchi (Corresponding author)

Reviewer#1

  1. The findings are presented well and relatively easy to understand, though the manuscript suffers from a high number of typographical errors and would benefit from additional proof reading. The topic is of high interest, and the results presented are also sufficiently novel and embedded in existing literature. It also appears to be mostly scientifically sound, though there are a few points concerning both style and content that should still be addressed:

Reply 1: We have carefully revised our manuscript for removing typographical and grammatical errors.

  1. For the calculations a rock-salt ordering of Bi and La in the solid solution was chosen. What is the reason for this? Were similar calculations carried out for other ionic arrangements? Did the type of ordering have a notable influence on the results? There should at least be a statement on this point included in the manuscript.

Reply 2: We have added Fig. S10 showing the DFT results of the rhombohedral cell (x = 1/2) with an ordered Bi and La configuration along the polar c axis. The ordered Bi-La occupation leads to an unrealistic metallic behavior because of the presence of the Fermi level inside Fe-3d-derived states; this is inconsistent with experimental results 1),2). The similar results were obtained for the tetragonal cell (x = 0.5) with an ordered Bi and La arrangement along the c axis. We have added the following sentences regarding these DFT results and the reason why we have adopted the rock-salt Bi-La ordering.

In page 4 in ‘2. DFT calculations’

For building a BiFeO3–LaFeO3 solid solution cell, there exists several choices of the arrangement of Bi and La. We adopted the rock-salt arrangement of Bi and La, especially along the polar c axis, as much as possible, as can be seen in Fig. S1 and S2. This is because the electronic structure in the rhombohedral cell with a layered Bi and La ordering along the polar c axis has a non-realistic metallic feature (Fig. S10), which is not consistent with the experimental fact of an insulating nature of BiFeO3–LaFeO3 solid solutions[46,47]. We therefore avoid such a Bi-La ordering along the specific crystallographic axis and adopt the rock-salt like orderings of Bi and La.

  1. The blue circle marking the relevant state in Fig. 3 i) is very hard to make out. In the other figures, it was additionally marked with an arrow; could this be added in Fig. 3 too?

Reply 3: We have added the corresponding arrow in Fig. 3i easy to see the blue circle.

  1. What is the point in showing all the bands in Fig. 3 h), 6 g) and 7 f)? Due to the high density of the lines, it is not possible to see any details. What information is the reader supposed to draw from this? Could these parts of the figure be left out?

Reply 4: As pointed out by Referee#1, the band structures in the valence band have many lines and then are not easy to see the details of the band dispersions. On the other hand, it is considered that a part of the band structures only in a limited energy range provide insufficient information on the entire electronic structures. By seeing the band structures not only in the whole valence band but also in the limited energy range including the Bloch function, we can recognize that the band (Bloch function) exists ca. 9th from the bottom of the valence band (in Fig. 3), which enable us to speculate that the corresponding band is present at the lowest energy among the bands containing the Bi-6p component. Therefore, we provide the band structures in the entire valence band as well as in the limited range including the Bloch function.

  1. The words ‘Bonding state’, ‘Antibonding state’ etc. in Fig. 5 are very hard to make out. Could this be presented in a larger font?

Reply 5: We have enlarged the size of ‘Antibonding state’ and ‘Bonding state’ in Fig. 5.

  1. The results show that the rhombohedral phase is the only stable phase across the entire composition range, and the tetragonal phase is energetically the most unfavorable in the unstressed state. As the authors point out, mechanical stress would stabilize the tetragonal phase. For the real-world understanding of the behavior of BLFO, it would therefore be interesting to know how this stress influences the results concerning the origin of ferroelectricity. Can it be assumed that the information on the hybridization of Bi_6p and O_2p being a major carrier of the ferroelectric polarization still holds true in the stressed system?

Reply 6: As described below, the origin of ferroelectricity in the tetragonal system remains unchanged when an in-plane compressive strain is chemically reduced by the La substitution. Compared with the BiFeO3 cell, an in-plane compressive strain decreases by 0.9 % for x = 2/8 and 4.8 % for x = 3/8 (these values are easily calculated from supplementary tables). Although the values of Ps and c/a show discontinuous drops at 2/8< x < 3/8, the Bi_6p-O_2p hybridization plays a dominant role in ferroelectricity in the tetragonal system irrespective of x. We therefore think that the origin of ferroelectricity remains unchanged when an in-plane compressive strain is in the range of 0 % to −5 %.

  1. The authors note that the drop in polarization in the tetragonal phase between x = 0.25 and x = 0.375 is due to the reduction of the number of Bi-chains through the structure. However, the threshold value for this should again depend on the Bi/La ordering. If the order is changed to allow for more Bi-chains at higher La content, does the threshold for the drop in polarization also shift?

Reply 7: As described in the response to Comment 2, the rhombohedral and tetragonal cells (x = 1/2) with the ordered Bi and La configuration along the polar c axis display an unrealistic metallic behavior as shown in Fig. S10; this result is inconsistent with experiments 1),2). Because the purpose of this paper is to elucidate the origin of Ps, we do not address the full choice of the configuration of Bi and La in the tetragonal system. We think that the overall dependence of Ps on x remains unchanged when we adopt a disordered distribution of a rock-salt like ordering of Bi and La.   

Reviewer#2

  1. Line 103-105 - why was rock salt arrangement selected out of the several existing choices the author mention ? Can the authors please specify what they mean by "as much as possible".

We have added the DFT results of the rhombohedral cell (x = 0.5) with an ordered Bi and La configuration along the polar c axis in Fig. S10. The ordered Bi-La occupation leads to an unrealistic metallic behavior because of the presence of the Fermi level inside Fe-3d-derived states. This is inconsistent with experimental results1),2). The similar results were obtained for the tetragonal cell (x = 0.5) with an ordered Bi and La arrangement along the c axis. We have added the following sentences regarding these DFT results and the reason why we have adopted the rock-salt Bi-La ordering:

In page 4 in ‘2. DFT calculations’

For building a BiFeO3–LaFeO3 solid solution cell, there exists several choices of the arrangement of Bi and La. We adopted the rock-salt arrangement of Bi and La, especially along the polar c axis, as much as possible, as can be seen in Fig. S1 and S2. This is because the electronic structure in the rhombohedral cell with a layered Bi and La ordering along the polar c axis has a non-realistic metallic feature (Fig. S10), which is not consistent with the experimental fact of an insulating nature of BiFeO3–LaFeO3 solid solutions[46,47]. We therefore avoid such a Bi-La ordering along the specific crystallographic axis and adopt the rock-salt like orderings of Bi and La.

Reviewer#3

  1. The introduction section of the manuscript seems insufficiently forming the background of the study. It discusses more the experimental background. Authors need to emphasize more the background on DFT calculations from the literature.

According to this reviewer’s comment, we have added the following sentences regarding experiments including some references:

In page 2 in ‘1. Introduction’

Rusakov et al. reported that single-phase materials with R3c symmetry can be prepared after annealing for composition 0 ≤ x ≤ 0.1 and the Pnma phase is stable at 0.50 ≤ x ≤ 1; these results were verified by synchrotron radiation X-ray diffraction, electron diffraction and high-resolution transmission electron microscopy[38]. They also found that an incommensurate phase in orthorhombic Imma symmetry is formed at 0.19 ≤ x ≤ 0.30. Karpinsky et al.[29] proposed a temperature-composition phase diagram, in which the ferroelectric R3c (x < 0.15) and the paraelectric Pnma (x > 0.4) phases are mediated through a bridging anti-polar phase in orthorhombic Pbam symmetry.

Moreover, we have added the following sentences regarding DFT calculations including some references:

In page 2 in ‘1. Introduction’

One possible reason for the wide variety of experimental reports on the phase diagrams is incomplete solubility of La on the A site[39]. Moreover, the above mentioned phases are energetically competing with each other and some of them are likely to be energetically degenerate.[31] Therefore, ab-initio studies based on density functional theory (DFT) are expected to provide clues for uncovering the ground-state crystal structure and the phase stability.

Lee et al. investigated the effect of the La doping on the variation of the off-center distortion and the orbital mixing in BiFeO3 by experiments in conjunction with DFT calculations[30]. They reported that both an Fe-O bond anisotropy and off-center cation displacements are suppressed by the La doping. As a result, the degree of Fe 3d-4p orbital mixing decreases in the solid solution samples. An impact of the La content on the polarization and the electronic band structure was also reported by You et al. They reported that the La doping induces a chemically driven rotational instability. It modifies the local crystal field along with the electronic structure, which gives rise to a direct-to-indirect transition of the bandgap and provides an enhancement in ferroelectric photovoltaic effect. In contrast, Tan et al. reported that the La doping has little influence on Ps in tetragonal BiFeO3. [34] In spite of extensive researches by DFT studies[30–35], the Ps evolution with the La content and its electronic origin still remain unclear.

  1. The Hubbard parameter of 6 eV was considered for the Fe-3d orbital in the study. How did the authors obtain this value? Authors should provide a reference.

In the literatures published before3)–6), the value of U−J for Fe-3d is in the range of 2–6 eV for BiFeO3 The value of band gap is enlarged when the U−J is increased, while the essential feature such as Ps and the valence-band electronic structure remains unchanged. One main reason why we adopted U−J = 6 eV is as follows: the bandgap becomes narrow for a specific Bi-La arrangement on the A site and eventually vanishes when the arrangement of Bi and La is an ordered configuration along the polar c axis. In order to maintain the band gap above ca. 2 eV, we set U−J to 6 eV for Fe-3d throughout our manuscript. 

We have added the following sentences:

In page 3 in ‘2. DFT calculations’

The on-site Coulomb interaction parameters of U−J for Fe-3d has been employed in the range of 2–6 eV for BiFeO3 [34,45–47]. The bandgap value is enlarged when U−J is increased, while the essential feature such as Ps and the valence-band electronic structure remain unchanged. One main reason why we adopted U−J = 6 eV is as follows: the bandgap becomes narrow for a specific Bi-La arrangement on the A site and eventually vanishes when the arrangement of Bi and La is an ordered configuration along the polar c axis, as will be described later. In order to maintain the band gap above ca. 2 eV, we set U−J to 6 eV for Fe-3d throughout the calculations.

  1. There are many typo errors (e.g. line no. 2, 92, 93, 111 etc.) in the manuscript. Please correct those.

We have revised our manuscript regarding typo errors in addition to grammatical mistakes.

1)       S. K. Singh and H. Ishiwara, Jpn. J. Appl. Phys. 45 [4B], 3194 (2006).

2)       S. Jangid, S. K. Barbar, I. Bala and M. Roy, Phys. B Condens. Matter 407 [18], 3694 (2012).

3)       J. B. Neaton, C. Ederer, U. V. Waghmare, N. A. Spaldin and K. M. Rabe, Phys. Rev. B 71 [1], 014113 (2005).

4)       Y. Wang, J. E. Saal, P. Wu, J. Wang, S. Shang, Z.-K. Liu and L.-Q. Chen, Acta Mater. 59 [10], 4229 (2011).

5)       L. You, F. Zheng, L. Fang, Y. Zhou, L. Z. Tan, Z. Zhang, G. Ma, D. Schmidt, A. Rusydi, L. Wang, L. Chang, A. M. Rappe and J. Wang, Sci. Adv. [ DOI:10.1126/sciadv.aat3438].

6)       Q. Tan, Q. Wang and Y. Liu, Materials (Basel). 11 [6], 985 (2018).

Reviewer 2 Report

The authors have investigated the origin of ferroelectricity in the BiFeO3-LaFeO3 system by ab-initio DFT calculations. The authors have acknowledged that ref - 30-35 to have studied this system but not able to confirm the origin of ferroelectricity (FE) clearly. This research article is interesting, very well written and a good read.

The reviewer has a few questions -

1. Line 103-105 - why was rock salt arrangement selected out of the several existing choices the author mention ? Can the authors please specify what they mean by "as much as possible".

The authors conclude that the Bloch function arising  from the indirect Bi hybridization mediated through O_2p is responsible for the origin of spontaneous polarization in the system. This is a new finding not clearly elucidated in the previous reported articles (30-35).

The reviewer therefore recommends publishing this article.

Author Response

(The authors gave the same response as above.)

Reviewer 3 Report

The authors have investigated the origin of the ferroelectricity in Bi1-xLaxFeO3 structure using density functional theory calculations. The manuscript provides a comprehensive study of the electronic and ferroelectric insights of the structure. Although there are few minor corrections which I have addressed below. I recommend this manuscript for publication after minor correction.

Comments:

1.       The introduction section of the manuscript seems insufficiently forming the background of the study. It discusses more the experimental background. Authors need to emphasize more the background on DFT calculations from the literature.

2.       The Hubbard parameter of 6 eV was considered for the Fe-3d orbital in the study. How did the authors obtain this value? Authors should provide a reference.

3.       There are many typo errors (e.g. line no. 2, 92, 93, 111 etc.) in the manuscript. Please correct those.

Author Response

(The authors gave the same response as above.)

Round 2

Reviewer 1 Report

With their revision and the answers in their reply letter, the authors have satisfactorily addressed most points raised in my previous report. A minor exception is point 3: contrary to the authors' statement, there is no arrow pointing out the blue circle in Fig. 3i) reproduced in my copy of the revised manuscript. Is there a problem with my display, or did the authors keep the earlier version of the figure on the submission server? In any case, this very minor technicality should not delay the publication of a manuscript that is otherwise of high quality and seems very suitable for publication.